# Conformational Insight on WT- and Mutated-EGFR Receptor Activation and Inhibition by Epigallocatechin-3-Gallate: Over a Rational Basis for the Design of Selective Non-Small-Cell Lung Anticancer Agents

**DOI:** 10.3390/ijms21051721

**Published:** 2020-03-03

**Authors:** Cristina Minnelli, Emiliano Laudadio, Giovanna Mobbili, Roberta Galeazzi

**Affiliations:** 1Department of Life and Environmental Sciences, Polytechnic University of Marche, 60131 Ancona, Italy; c.minnelli@staff.univpm.it (C.M.); g.mobbili@staff.univpm.it (G.M.); 2Department of Science and Engineering of Matter, Environment and Urban Planning Polytechnic University of Marche, 60131 Ancona, Italy; e.laudadio@staff.univpm.it

**Keywords:** EGFR, Epigallocatechin-3-gallate, molecular docking, non-small cell lung cancer

## Abstract

Non-small cell lung cancer (NSCLC) represents a difficult condition to treat, due to epidermal growth factor receptor (EGFR) kinase domain mutations, which lead to ligand-independent phosphorylation. Deletion of five amino acids (ELREA) in exon 19 and mutational change from leucine to arginine at position 858 (L858R) are responsible for tyrosine kinase domain aberrant activation. These two common types of EGFR-mutated forms are clinically associated with high response with Tyrosine Kinase Inhibitors (TKI); however, the secondary T790M mutation within the Tyrosine Kinase Domain (TKD) determines a resistance to these EGFR-TKIs. Using molecular dynamic simulation (MD), the present study investigated the architectural changes of wild-type and mutants EGFR’s kinase domains in order to detect any conformational differences that could be associated with a constitutively activated state and thus to evaluate the differences between the wild-type and its mutated forms. In addition, in order to evaluate to which extent the EGFR mutations affect its inhibition, Epigallocatechin 3-Gallate (EGCG) and Erlotinib (Erl), known EGFR-TKI, were included in our study. Their binding modes with the EGFR-TK domain were elucidated and the binding differences between EGFR wild-type and the mutated forms were evidenced. The aminoacids mutations directly influence the binding affinity of these two inhibitors, resulting in a different efficacy of Erl and EGCG inhibition. In particular, for the T790M/L858R EGFR, the binding modes of studied inhibitors were compromised by aminoacidic substitution confirming the experimental findings. These results may be useful for novel drug design strategies targeting the dimerization domain of the EGFR mutated forms, thus preventing receptor activation.

## 1. Introduction

Lung cancer is the most common form of tumor with the highest incidence in the world, representing approximately 75% of lung cancer cases [1]. Surgery is the conventional treatment of early non-small-cell lung cancer (NSCLC), but only 20–30% of patients are diagnosed at resettable stage (stage I–II). NSCLC treatment has changed with the development of targeted therapies based on the design of biological agents targeting critical pathways in lung cancer cells. In particular, several benefits arise with the use of tyrosine kinase inhibitors (TKIs), targeting epidermal growth factor receptor (EGFR), the activation of which is involved in cell proliferation, blocking apoptosis, angiogenesis and migration [2]. The phases of the activation are reported (Figure 1): the receptor binds to a protein called Epidermal Growth Factor (EGF) via the extracellular domain, leading to the activation of the receptor, which reorients the extracellular and intracellular domains allowing the homodimerization with another EGFR receptor, or heterodimerization with other proteins of the Epithelial Growth Factor Receptor family (Figure 1) [3,4]. Homo-heterodimerization induces the autophosphorylation of the intracytoplasmic EGFR tyrosine kinase domains and finally the recruitment of a series of proteins, which will quickly lead to cell differentiation and proliferation [5]. Meanwhile, aberrant EGFR signaling can lead to cancer transformation and activation of invasion and metastasis [6,7,8].

In this regard, several studies have tried to identify prognostic and predictive factors associated with sensitivity or resistance to anti-EGFR agents [9,10,11]. These agents are small molecules that act as competitive inhibitors with ATP, avoiding its phosphorylation interacting with the ATP binding site. This strategy allows to inhibit the EGFR activation, however the onset of specific mutations of the coding gene EGFR leads to changes in the some key aminoacids that are fundamental for binding of the inhibitor. These alterations often render canonical inhibition strategies ineffective. The most common drug-resistance mutations are the “in-frame” deletion that eliminate the five ELREA amino acids (corresponding to the Glu746-Ala750 fragment) in exon 19 and mutational changes from leucine to arginine in position 858 (L858R) [12,13]. These alterations are associated with the high response to the first-generation inhibitors. Despite the marked initial response, lung cancers inevitably acquire resistance to these TKIs most frequently owing to the secondary T790M mutation within the ATP site of the receptor (50–60% of resistant cases), which determines a lower the binding affinity of the TKIs to the ATP-kinase-binding pocket, comprising gefitinib and erlotinib [14,15,16]. Moreover, their high skin and gastrointestinal toxicity [17] leads to dose reductions or even cessation of treatment. Therefore, the assessment of the molecular basis of the inhibition is crucial to better understand the mechanisms of resistance to these agents and to identify new and less toxic TKIs. Among the natural anticancer agents, Epigallocatechin-3-gallate (EGCG), a major biologically active constituent of green tea, is known to produce several beneficial effects in human health [18]. It acts as inhibitor that competes for ATP binding site of EGFR cytoplasmic domain, in erlotinib-sensitive (wild type and ELREA deletion form) and -resistant cell lines in which the secondary T790M mutation is present (EGFR L858R-T790M) [19]. This antioxidant compound interacts with the double mutated EGFR form a little more effectively than erlotinib, thanks to a high number of hydrophilic groups, which allow it to create polar interactions with amino acid, even if it is not particularly active [20].

In this study, an in silico approach was used to shed light on the molecular basis of resistance mechanisms of EGFR-targeted therapies based on TKIs. To do this, three different EGFR forms, relative to wild type, mutated T790M/L858R and ELREA deletion forms were rebuilt starting from the X-ray crystallographic structure of the wild-type EGFR kinase domain (PDB ID 4HJO) [21]. We decided not to consider the single mutation L858R since there are no experimental data concerning ECGC activity; instead, the double mutation T90M-L858R EGFR was studied since it can help in understating the molecular mechanism of resistance to erlotinib.

Due to its importance during receptor activation, the dimerization domain full modeled for the first time; no computational data are available in literature about the role of mutations in the protein dimeric complex association corresponding to its active state. For this reason, the understanding of the dimerization domain conformational movements represents a crucial point to acquire knowledge for the EGFR activation blockade strategy. It has been suggested [22,23] that the binding of the ligand in the extracellular domain, conformational constrains it, inducing a rotation of the rigid intracellular TK domain and thus allowing the dimeric coupling; however, the more flexible D-domain must arrange in a suitable way to allow this matching, i.e., the transition from the inactivated to the activated form. It is also known that the double mutant T790M/L858R lies in a “costitutively activated” state as result from [24], when it heterodimerize with Her2. Thus, the MD investigations of the D-domain spatial orientation can help to explain this peculiar behavior, and the differences between the wild type and the mutated EGFR forms, that is at the basis of the common-therapy resistance. In addition, the binding modes of a known EGFR-TK inhibitor, erlotinib (Erl) and epigallocatechin-3-gallate (EGCG) that showed in vitro promising inhibitory activity to the three studied EGFR forms were analyzed and compared [25,26]. In fact, even if several studies investigated the molecular basis of EGCG inhibition for wild type EGFR cytoplasmic domain, the EGCG inhibition and the EGFR mutations of the TK domains have never been explained.

## 2. Results and Discussion

### 2.1. Wild Type, T790M/L858R and ELREA Deletion EGFR Differ for Conformational Mobility of the Dimerization Domain

The EGFR cytoplasmic wild type domain was completed modeling the dimerization domain; then relaxed and stabilized into its physiological environment (water and 0.15 M NaCl). To verify the reliability of the used building model approach (see methods section and SM material), the resulting wild type EGFR model was superimposed with the crystallized EGFR TK-domain (i.e., 4HJO pdb code), the Ramachandran plot analyzed, and in addition the structure was submitted to Verify3D analysis (https://servicesn.mbi.ucla.edu/Verify3D/), showing a high accuracy degree of the model obtained (see Appendix A). The same approach was used for the mutated T790M/L858R and ELREA deletion forms, and the differences in the conformational flexibility between the three EGFR forms were evidenced using structural analyzes’ tools. According to a widely accepted model [22,23], at the basis of the dimerization process, there is a rotation of the TK-domain induced by the extracellular EGF-binding. This TK domain is overall rigid whilst the D-domain gains more flexibility thanks to the 16aa long linking loop (see Appendix A). However, to allow the dimeric matching, this switching loop that links the TK- with the D-domain, must arrange in suitable way to allow the coupling i.e., putting the D-domain at interface of the activated dimer.

The EGFR TK-domains crystallized structures do not contain entirely this D-domain, thus its complete modeling and structural analysis can help to identify the spatial movements of the tyrosine kinase (TK) and dimerization (D) domains involved in the activation mechanism. Furthermore, it can shed light into the different behavior between the wt and the mutated form.

The proper orientation of the D-domain, that is important for inactive-active conformation switching, and the H-bond interactions involved within the N-lobe of the TK domain (Figure 2) can represent the main driving force in this spatial reorientation [27]. The active conformation requires that the αC-helix switch closer the ATP-binding site within the TK domain, to form hydrogen bond interactions with Ser768-Asp770 and Tyr827-Arg831 [28,29]. In the EGFR inactive conformation, instead, the αC-helix is positioned far away from the binding site, opening a large back hydrophobic pocket. This different αC-helix orientation indirectly influence also the D-domain spatial disposition.

Indeed, analyzing the MD trajectories, the αC-helix in the EGFR wild type model remained in the “inactive state” position through all the simulation time. On the contrary, in the T790M/L858R model, the formation of additional stable H-bonds involving Ser768, Asp770, Tyr827 and Arg831 residues was observed (Figure 3A), inducing the αC-helix bending to a permanent, thus constitutively active conformation (Figure 4). This is in agreement with already reported results in which the T790M/L858R TK-domain is considered always in the activated state [24].

ELREA deletion TK-domain conformational movements were also investigated, comparing them with stabilized structure of T790M/L858R mutated receptor. The aim was to evaluate the presence of possible similarities in the spatial arrangement of the domains, validating the hypothesis of a possible constitutive activation also for these mutated forms. No specific H-bond interactions involving αC-helix and N-lobe was observed for ELREA deletion model, but a very similar orientation of dimerization (D) domain was also found (Figure 3B and Figure 4). However, the lacking of specific H-bonding interactions induce a partial conformational switching between the closed (active) and open (inactive) orientations. This data means that there is no evidence of a constitutive activation in this receptor, but the deletion of five ELREA aminoacids influenced the spatial movements of TK-sub domains.

To confirm that the D-domain orientation can lead to the constitutive activation of T790M/L858R EGFR, or any of the other EGFR forms, a comparison of our models with the activated dimeric structure of another TK receptor (pdb code: 6ewx [30]) was carried out. This structure contains two TK domains in dimeric association, corresponding to the activated state. In particular, we used this model to compare our EGFR monomers and evaluate the capability of the three MD-stabilized EGFR forms to associate in a stable dimeric complex. To do this, a three dimensional fitting between 6ewx and EGFR in different form was carried out through a superimposition match, evaluating the root-mean-square deviation (RMSD) between 6ewx and EGFR structures. For the wild type model, the RMSD value obtained from superimposition with 6ewx was 1.01 Å. For wt EGFR, the orientation of the D-domain sterically prevented an efficient interaction between the two monomers (Figure 4A,B) along the MD trajectory, in which it retained always an inactive conformation (no constitutive activation, in agreement with literature data). On the contrary, as already anticipated in the previous paragraph, for T790M/L858R model an RMSD value of 0.87 Å respect 6ewx file was found, and after 80 ns of simulation the D-domain adopted a stable bended conformation, which allowed to two monomers to efficiently couple and form a stable dimer complex (Figure 4C,D). These results support the hypothesis that the specific movement of the Nlobe of the TK domain and the spatial bended orientation of the D-domain induce the constitutive activation of T790M/L858R EGFR receptor. A dimer complex was also observed for ELREA deletion form, in which an RMSD value of 0.98 Å respect 6ewx file was found, and a large part of the D-domain was oriented to allow a discrete amount of connections between aminoacids of the two monomers (Figure 4E,F). The substantial difference with respect to the T790M/L858R mutated form concerns the fact that albeit sterically possible, no highly specific interactions between amino acids were observed; this means that even if deletion favor a receptor conformational rearrangement in a closed form, the resulting activation is not so strong as for the double mutated receptor.

The key aminoacids involved in this activation mechanism were identified and the MD trajectories were further analyzed. The root-mean-square deviation values for all three EGFR models showed the achievement of a steady state, confirming that stable conformations were obtained (Figure 5A and see Appendix A). The last 20 ns of MD simulation were taken into account for root-means-square fluctuation (RMSF) profiles. The RMSF data showed a similar trend in different regions of EGFR cytoplasmic domain for all the three EGFR forms (Figure 5B). T790M/L858R EGFR model showed an increase in aminoacids fluctuation, in particular for the D-domain, with respect the others, which can be associated with constitutive activation above described.

Based on RMSF results, the movements of the D-domain of EGFR models were analyzed in detail, and the mean square displacement (MSD) of this domain from the set of initial positions was computed. Inside the simulation boxes (Figure 5C), the D-domain of the EGFR wild type model shifted by about 4 nm^2^ with respect to the initial position. Anyway, this slow movement suggests no peculiar difference respect to the beginning of the MD trajectory and keeps EGFR wild type model in the inactive form. Different results have been obtained for both double mutated and ELREA receptors, since after 50 ns, both the D-domains displacement drastically increased to 12 nm^2^ respect to the initial positions. These results evidenced that the differences in aa sequence related to the mutations onset strongly influence this domain spatial movements.

Finally, we focused on the radius of gyration (Rg) analysis that is commonly used to investigate the compactness degree of EGFR structures. The Rg results (Figure 5D) showed that the wild type EGFR domains reached approximately 2.7 nm after 80 ns of MD simulation. For EGFR-ELREA deletion, the Rg value reached to 2.5 nm at the end of MD simulation. The compactness of these two models appeared similar, while a different behavior was observed for T790M/L858R EGFR. The starting Rg value was comparable to the other two models (around 2.5 nm), but after 80 ns of MD simulation it drastically decreased to 1.74 nm. Overall, the T790M/L858R EGFR-TK model results much more compact respect to the other two models, a compactness that can be ascribed to the adoption of a fully bended activated conformation in which the D-domain much lay close to the NLobe, in a correct orientation to promote an efficient dimerization.

### 2.2. Known Inhibitors of EGFR-TK Domains Bind Differently with the Three EGCFR Receptors

The binding modes of Erl and EGCG were investigated together with their binding affinity for the three EGFR forms. Both Erl and EGCG are reported to bind the ATP binding pocket of EGFR, despite having a much different chemical structure. Then, the docked EGFR-inhibitor complexes were relaxed by MD simulation, to identify eventual ligands reorientations inside the cleft. The final structures were analyzed in order to better understand the key residues involved in the inhibition mechanism of these ATP competitive inhibitors.

The presence in the ATP binding pocket of a high number of purely apolar residues makes the hydrophobic interactions the main binding force for the stabilization of the Michaelis complex; however, the presence of polar groups increases the affinity and specificity of these ATP competitive inhibitors. Both Erlotinib and Epigallocatechin-3-gallate share two opposite chemical physical features (hydrophobic/polar moieties) that fully characterize them as “typical” EGFR competitive inhibitor of ATP binding site (Figure 6). The summary of binding energy obtained with the dynamic molecular docking approach is shown in Figure 6A.

Then, in order to evaluate the free energy of binding with higher accuracy, MD simulation for each EGFR-Ligand complex was carried out, and an MM/PBSA (Poisson Boltzmann Surface Area) calculation [31] on the last 5 ns of the production runs was performed (Figure 6B). The results showed very good stability for the complexes of Erl and EGCG with wild type and ELREA deletion EGFR forms. In line with experimental data, it is well established the ability of Erlotinib to target TK of EGFR-ELREA [32], yet a significant benefit of Erl treatments was found also in patients with EGFR-wild type [33,34]. In addition, EGCG shows high growth inhibition activity towards EGFR-wt and ELREA harboring NLSC cell lines [35]. On the contrary, the MD trajectories did not show the formation of stable complexes between Erl or EGCG and the T790M/L858R mutant, and in fact a drastic decrease in the binding affinity for both ligands was evidenced (Figure 6B). This result is in line with experimental observation, since this double mutated receptor is resistant to Erlotinib-chemotherapy [36,37]. In the following sections a detail of inhibition mechanisms in these three EGFR forms were shown.

#### 2.2.1. Erl and EGCG Binding Mode with EGFR Wild Type

The binding mode of Erl with EGFR wild type model was studied. In line with in vitro experimental data on the Erl-EGFR affinity (*K_i_* = 0.1–0.4 μM) [38], the in silico results revealed that Erl bound to EGFR with high affinity. Only one binding pose for Erl has been identified, corresponding to that co-crystallized in the ATP binding site (pdb code 4HJO) [39] (Figure 7B). The binding energy and inhibition constant (*K_i_*) calculated for the best scored pose were −7.3 kcal/mol (E_MM/PBSA_ −71 kcal/moL) and 0.87 μM, respectively. The specific interactions of Erl with residues of the wild type EGFR-TK domain are reported in Figure 7A, and an important hydrophobic contribution was found. The main apolar character of Erl determines a strong binding affinity with the apolar aminoacids found in the ATP binding site, and these apolar interactions were retained for all MD simulation (Appendix A), corresponding to highly negative free Gibbs energy values (Figure 6).

Analyzing the EGCG binding with the wild type EGFR, one preferential and favorite binding pose was found, involving directly the ATP binding pocket (Figure 7A) in agreement with experimental results. Due to the high number of OH groups of this catechin, five different Hbond interactions were found, involving Ile744, Gly796, Asn842 and Asp855 residues, and in particular with Cys797, which is very important for the EGFR inhibition as already demonstrated in other studies [40]. In addition, further hydrophobic interactions with different aminoacid residues were observed (Figure 7B). For these reasons, the calculated EGCG binding energy and *K_i_* values (−9.0 kcal/moL (E_MM/PBSA_ −98 kcal/moL) and 0.13 μM, respectively) were much lower than those obtained for Erl. MD simulation showed no reorientation of EGCG in the ATP binding site (Appendix A), meaning that the complex is very stable and energetically favored (Figure 6).

#### 2.2.2. Erl and EGCG Binding with EGFR ELREA Deletion

Erl binding with EGFR ELREA mutated form was evaluated. In this receptor, the variation consists in the deletion of E146-A150 aminoacidic sequence of the ATP binding site. Once again, only one preferential binding pose was found, within the ATP binding pocket, and an increase of binding affinity of Erl with respect to that predicted for the same compound and the wild type EGFR, was observed (E_MM/PBSA_ −97 vs. −71 kcal/mol) (Figure 8A). In this complex, Erl made three additional H-bonds interactions Val730, Ala747 and Lys749 (Figure 8B). A higher hydrophobic contribution was found, due to a more deeply located binding pose inside the ATP binding site with respect to that observed in the wild type Erlotinib-EGFR complex. This contribution determined an increase of binding affinity and suggested an increasing Erlotinib inhibitory activity in presence of ELREA EGFR receptor, in agreement with experiment observations [33,34,35,36,37,38]. MD simulation trajectories revealed that this complex is very stable (Appendix A) as confirmed by MM/PBSA binding energy values (Figure 6).

The EGCG interactions in relation to the ELREA EGFR receptor were then evaluated. In addition, in this case, since some residues within the ATP binding site are missing, different interactions with respect to the previous system were observed. In particular, EGCG made six H-bond interactions involving Ser724, Ala747, Gln788, Met790, Cys794, Arg838 residues. In this case, the hydrophobic contribution was not so relevant (Figure 8B). The calculated binding affinity and *K_i_* of EGCG in association with the ELREA deletion EGFR model was −7.45 kcal/moL (E_MM/PBSA_ −75 kcal/mol) and 0.84 μM, respectively, showing a good affinity, even if not as good as for EGCG-wild type EGFR complex (E_MM/PBSA_ −75 vs. −97 kcal/mol). MM/PBSA calculations showed for EGCG a lower affinity in this case with respect to Erl (Appendix A, Figure 6).

#### 2.2.3. Erl and EGCG Binding with EGFR T790M/L858R

As described, this type of mutation increases the EGFR receptor affinity for ATP [21]. The binding of both Erl and EGCG with double mutated T790M/L858R receptor was investigated and Molecular docking showed that the best binding pose of Erl was found one again inside the ATP cleft (Figure 9A). Some spatial differences comparing with other two Erl-EGFR complexes were found, and different aminoacids were involved in the binding of Erl (Figure 9B): an H-bond interaction with Cys797 and an important hydrophobic interaction with Met793. In addition, the bulkier Met790 side chain imposed steric strains to Erl, thus decreasing its affinity. This was confirmed by the binding energy and the calculated K_i_ obtained (−6.5 kcal/mol (E_MM/PBSA_ −39 kcal/mol) and 154.3 μM respectively). MD simulation put into evidence a reorientation of Erl in binding cleft, but no stable complex was observed (Appendix A), justifying the low binding affinity obtained (Figure 6). These data support the clinical evidence of an inefficient inhibition activity of Erl for T790M/L858R EGFR system [36,37].

Focusing on EGCG, the preferential binding site corresponding to the ATP binding pocket was confirmed (Figure 9A), but also for this compound, a different binding mode respect to that characterized for the wild type and ELREA deletion receptors was observed. EGCG makes four H-bonds interactions involving Tyr731, Ile744, Met793 and Cys797 (Figure 9B), indicating a relevant contribution of polar interactions in the complex stabilization as expected. Despite this observation, a significant decrease in the binding affinity value was observed (E = −6.92 kcal/mol (E_MM/PBSA_ −54 kcal/mol); *K_i_* = 134.11 μM) with respect to the other two EGFR-EGCG complexes. This result can be justified on the basis on a lower hydrophobic contribution associated with the mutation of residues 790 (Thr to Met) and 858 (Leu to Arg) that were involved in the previous binding. MD simulation revealed the formation of little more stable complex respect to the Erl-T790M/L858R EGFR one (Appendix A), but the energy contribution is much far in affinity with respect that calculated for the wild type and ELREA EGFR complexes (Figure 6). Nonetheless, the overcome resistance in EGFR T790M/L858R after treatment with a combination of EGCG and TKIs was recently found [20].

## 3. Materials and Methods

### 3.1. EGFR TK Cytoplasmic Domain Modeling

EGFR is a 1197 aa long protein kinase receptor with three main domains: The N terminal domain is the extracellular one, which is involved in antibody therapy for cancer insurgent [21,41]. The second domain is a simple α-helix which transverses the cell membrane, while the C terminus domain is the cytoplasmic one, composed by tyrosine kinase (TK) and dimerization (D) domains. The TK-domain is 265 aa long, and it was yet crystallized, corresponding PDB 4HJO structure [39]. This pdb file was used, as starting point to rebuilt the entire cytoplasmic domain. The dimerization domain is 230 aa long, and it was obtained using I-TASSER (https://zhanglab.ccmb.med.umich.edu/I-TASSER/, accessed on February 2019) through a building block approach, because no complete crystallographic data are available for this domain [42]. To obtain a valid 3D model, we rebuilt the D domain considering core building blocks using pdb existing templates (4rj4 and 5cnn, that covers the first 80 residues) that shared 100% identity and reconstructing the lacking residues in a 20 aa long fragment a time using I-TASSER fragment libraries. The junctions between carboxyl and amino terminal residues of each block were modeled using GalaxyWEB (http://galaxy.seoklab.org; accessed on February-March 2019) [43] sculpting tools implemented in Schrödinger suite (Schrödinger Release 2017-1: Schrödinger, LLC, New York, NY) [44]. DISULFIND online server [45] was used to predict the disulfide bonding state of cysteines and their disulfide connectivity. No disulfide bridges were found in the intracellular EGFR domain. 

Modloop webserver (https://modbase.compbio.ucsf.edu/modloop/, accessed on February-March 2019) [46] was used to predict the loop conformations considering spatial restraints and without considering the databases of known protein structures. The completely rebuilt EGFR domains of wild type, T790M/L858R and ELREA deletion EGFR forms were taken into account for the *in silico* investigations. Preparation for 100 ns productive MD simulation on the three different EGFR cytoplasmic domain models was carried out with different steps of minimization and annealing protocol with AMBER99SB-ildn force field. The simulation box to 120 × 120 × 120 Å^3^ containing wild type EGFR structure was settled, and 54867 TIP3P water molecules, were added to solved it. 169 Na^+^ ions and 156 Cl^−^ ions were used to reach the physiological considering the net charge of EGFR protein. Solvation and ions addition were all completed with GROMACS 5.0.4 (http://www.gromacs.org/) [47,48]. A total of 10,000 cycles of steepest descent energy minimization, followed by 5000 cycles of conjugate gradient minimization were sufficient for the maximum force to converge to the energy threshold of 1000 KJ/mol/nm. The following annealing steps were conceived to let the protein to gradually accommodate in the salt-aqueous environment. In all runs, Verlet cutoff [49], combined with PME for electrostatics [50] was applied. The cutoff for the calculation of the Van der Waals force was set to 1.2 nm, with the force smoothly was switched to zero between 1.0 and 1.2 nm. Atom velocities were first generated at 310 K in the NVT ensemble, using the Maxwell distribution function with generated random seed, and a weak temperature coupling using the Berendsen thermostat. Time constant of 1 ps was applied to maintain the reference temperature (310 K) for the whole run. After annealing simulation run of 2 ns, a switch to the NPT ensemble was made, maintaining the weak coupling also for pressure control (i.e., Berendsen barostat). For all simulation runs, the isotropic conditions were set with a reference pressure of 1 atm and a time constant for coupling of 5 ps. Position restraints were applied to EGFR protein. A shift to the Nosé-Hoover [51,52] and Parrinello-Rahman algorithm for pressure coupling [53,54] was operate for production phase in NPT Ensemble. Then, a 100 ns-long dynamic simulation was run for each system, implementing an accurate leapfrog algorithm or interacting Newton’s equations of motion with a time step of 0.002 ps. This already validated MD protocol [55,56] was used for wild type, T790M/L858R and ELREA Deletion EGFR forms, while the analysis of the simulations’ trajectories was performed by means of the VMD [57] and CHIMERA software [58].

### 3.2. Inhibitors Docking to EGCFR Receptors

The ligands-EGFR interactions were investigated in order to rationalize the behavior of Erl and EGCG molecules, starting from a blind docking procedure for three selected EGFR structures with AutoDock Suite 4.2 [59]. In order to assess the correct position of the ligands, a full flexible focused docking centered on the same site was performed. Its graphical front-end, AUTODOCKTOOLS [60] was used to add polar hydrogen atoms and partial charges for proteins, while ligand charges were obtained at AM1 level and added manually. Atomic solvation parameters and fragmental volumes for the proteins were assigned using the ADDSOL tool (included in the program package). Flexible torsions in the ligands were assigned with the AUTOTORS module and all dihedral angles were allowed to rotate freely. Affinity grid fields were generated using the auxiliary program AUTOGRID. A grid field of 126 × 126 × 126 Å^3^ cube was used with grid points separated by 0.34 Å centered on the best scored conformation obtained. Lennard Jones parameters 12_10 and 12_6 (supplied with the program package) were used for modeling H-bonds and van der Waals interactions, respectively. Starting from the best docked poses obtained with blind docking approach, a refined focused docking was carried out using a Lamarckian genetic algorithm with the flexible ligands and the rigid EGFR receptors, with a population size of 300, 10,000,000 evaluations, and a maximum of 27,000 generations for 100 GA runs. The grid field was a 70 × 70 × 70 Å^3^ cube, and the resulting docked conformations were clustered into families of similar binding modes, with a root mean square deviation (RMSD) clustering tolerance of 2 Å. The lowest docking-energy conformations were considered as the most stable orientations. The docking energy represents the sum of the intermolecular energy and the internal energy of the ligand while the free-binding energy is the sum of the intermolecular energy and the torsional free energy [61]. The docked complexes binding energies were calculated by an empirical free energy force field with a Lamarckian genetic algorithm (LGA), which provides a fast prediction of conformation and free energy. This calculated free binding energy can be related to the inhibition constant (*K_i_*) through the known thermodynamic law ΔG= −RT ln*K_i_*.

The computational protocol combining Blind/focused docking in triplicate [62,63] was applied for both Erl and EGCG in association with wild type, T790M/L858R and ELREA deletion-EGFR-receptor; the most stable EGFR-ligand complexes for each couple ligand-receptro were relaxed by 20 ns of MD simulation following the MD protocol previously described. Finally, to the stability and the interaction of Erl and EGCG bound with three different EGFR forms are evaluated and compared.

Free Gibbs Binding energy was calculated with the MM-PBSA method (Molecular Mechanics/Poisson Boltzmann Surface Area) using g_mmpbsa tool [31] with default settings. The last 10 ns of production run in the simulations were used and snapshots were extracted every 10 ps and energetic terms calculated. Results are in terms of average and standard deviations for all energetic components.

## 4. Conclusions

EGFR-TK domain represents a valid target for anticancer therapies. In this work, complete cytoplasmic models relative to the wild type, T790M/L858R and ELREA deletion forms were modelled consisting of the tyrosine kinase and dimerization domains. The fluctuations of aminoacids and the relative displacement of the two cytoplasmic domains (TK- and D-domains) were investigated at the molecular level using atomistic molecular dynamics. The residues mutations influence the spatial movements and the stabilizations of the three simulated receptors. In particular, the main role of the dimerization domain was evidenced identifying the active-inactive conformational switching. Moreover, the T790M/L858R EGFR showed the permanent switch to a constitutive active form along MD simulation, thus in a form always ready for dimerization coupling. The identified conformational movements of the EGFR domains can compromise the canonical strategies to inhibit the EGFR activation, opening the way to new molecular drug design approaches able to avoid the critical problem of the emerging resistance mechanisms due to EGFR mutations.

In addition, the studies on the interactions of two known inhibitors with EGFR different built models evidenced the molecular basis of inhibition. The main forces on the binding modes of Erl (a synthetic inhibitor) and EGCG (a natural compound aging as EGFR inhibitor) were studied to understand the way to keep EGFR in its inactive form. The aminoacids mutations directly influence the binding affinity of these two inhibitors, resulting in a different efficacy of Erl and EGCG inhibition. In particular, for the T790M/L858R EGFR, the binding modes of studied inhibitors were compromised by aminoacidic substitution confirming the experimental findings. Thus, a novel strategy could be the drug design targeting the dimerization domain of the EGFR mutated forms, avoiding receptor activation and bypassing the study of mutations in the receptor tyrosine kinase domain.

A drug design approach aimed to block the mutated EGFR domains activation are currently underway in our laboratory, associating the in silico prediction with in vitro/in vivo experiments to confirm the inhibitory mechanisms against the different EGFR forms and to gain more insights on the evolution and biochemical implication of the molecular structure of the complexes.

## Figures and Tables

**Figure 1 ijms-21-01721-f001:**
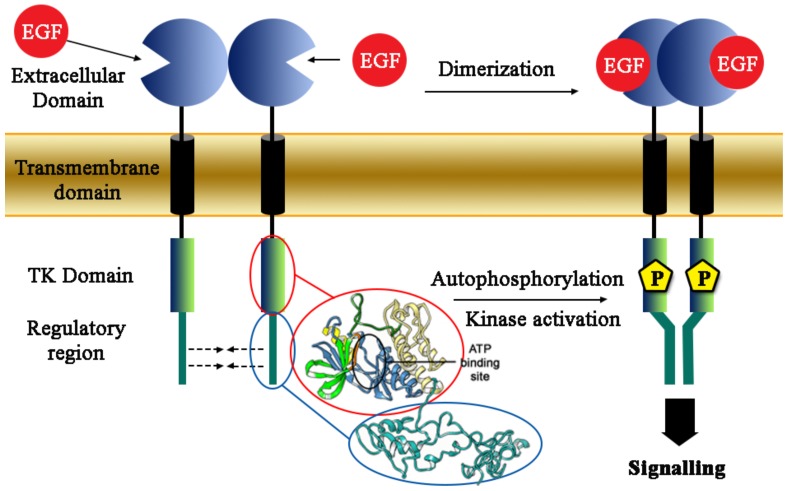
Epidermal growth factor receptor (EGFR) structure in cellular membrane and the dimerization mechanism.

**Figure 2 ijms-21-01721-f002:**
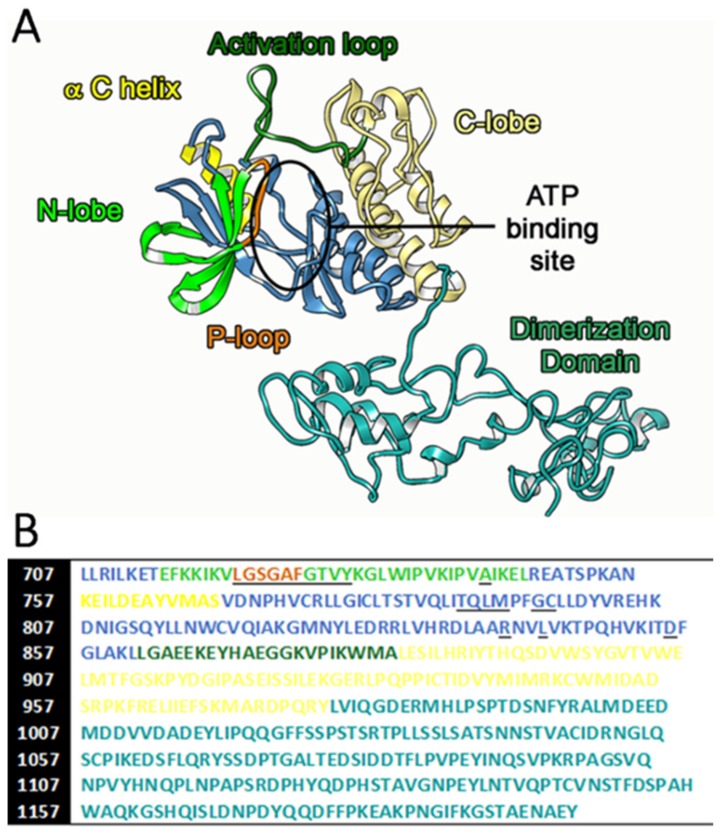
The complete structure of wild type EGFR cytoplasmic tyrosine kinase (TK) domain after molecular dynamic (MD) simulation (**A**) and the corresponding FASTA sequence (**B**). For clarity, we reported the same color index on both figures, while the underlined aminoacids make up the ATP binding site.

**Figure 3 ijms-21-01721-f003:**
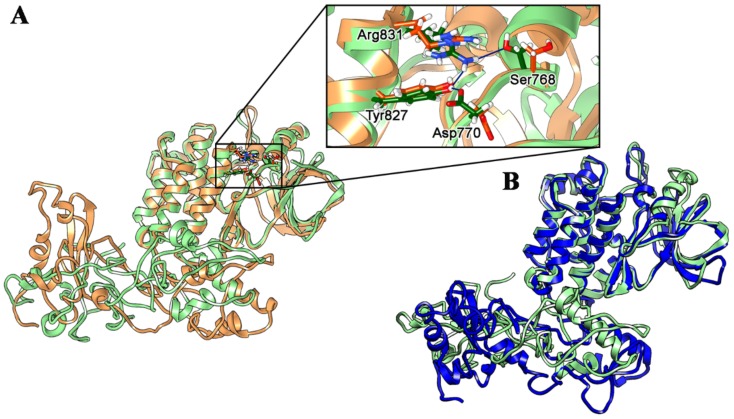
H-bond interactions of Ser768, Asp770, Tyr827 and Arg831 residues in the T790M/L858R EGFR domain. We reported the structure at the beginning (orange) and at the end (green) of MD simulation (**A**). Superimposition of T790M/L858R (green) and ELREA deletion (blue) EGFR structures in the corresponding closed conformations (**B**).

**Figure 4 ijms-21-01721-f004:**
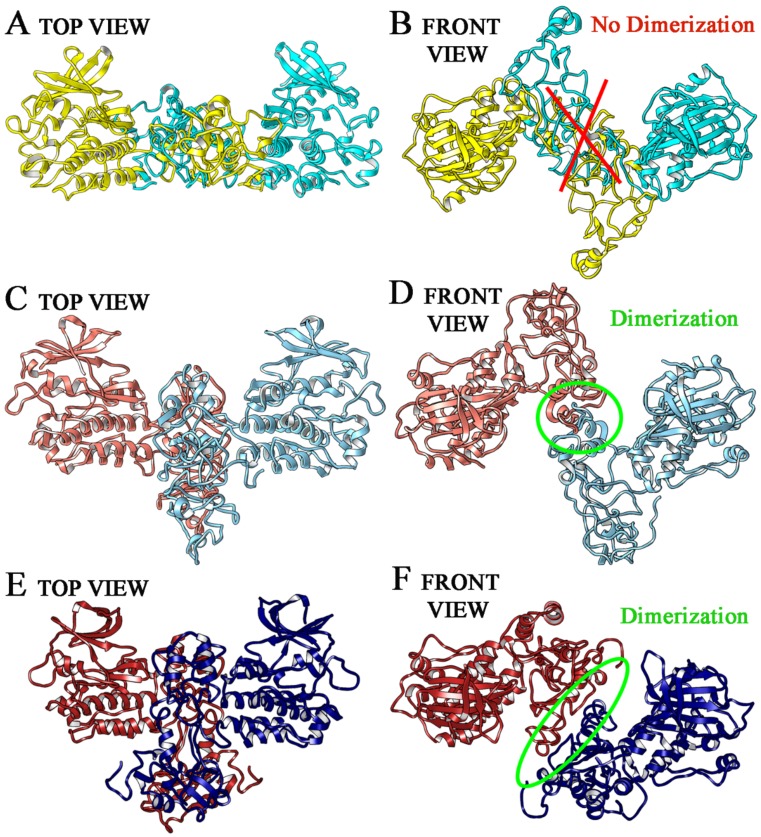
EGFR cytoplasmic domains in dimerization form. We reported in yellow and light green the domains of wild type EGFR (**A top view**,**B front view**), in orange and light blue the T790M/L858R EGFR (**C top view**,**D front view**) and in red and blue the ELREA deletion EGFR (**E top view**,**F front view**).

**Figure 5 ijms-21-01721-f005:**
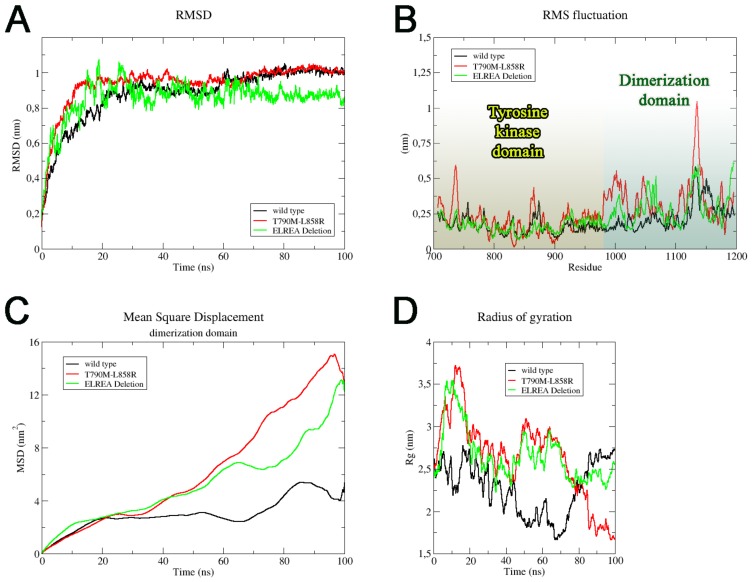
Structural analysis of EGFR cytoplasmic different forms. We reported root-mean-square deviation (RMSD) (**A**), root-means-square fluctuation (RMSF) (**B**), mean square displacement (MSD) of the dimerization domain (**C**) and radius of gyration (Rg) (**D**), For all graphs we reported the wild type, the T790M/L858R and ELREA deletion forms (black, red and green, respectively).

**Figure 6 ijms-21-01721-f006:**
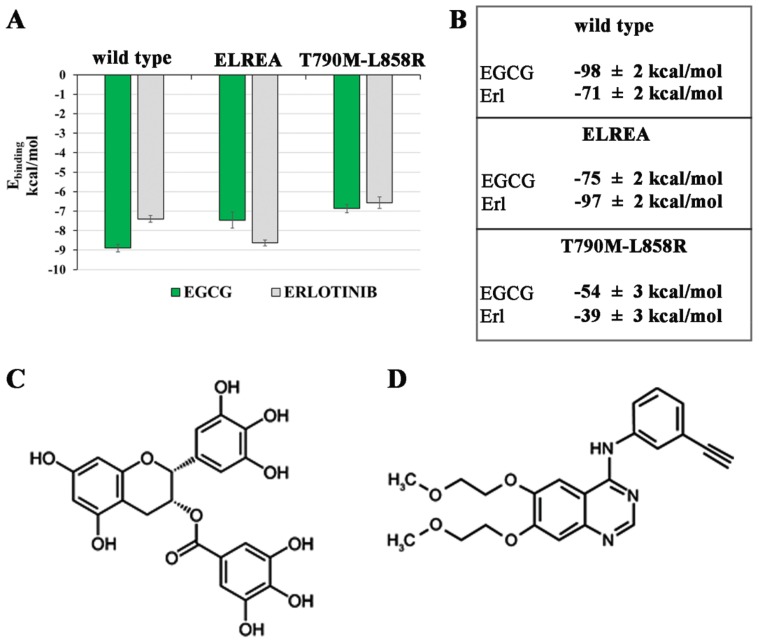
Binding energy of Epigallocatechin-3-gallate (EGCG) and Erlotinib in association of wild type, ELREA deletion and T790M/L858R mutated EGFR forms obtained with the molecular docking (**A**) and molecular dynamics approach (**B**). We reported the total MM/PBSA energy calculated in the last 5 ns of MD simulation in kcal/mol.; 2D structure of EGCG (**C**) and Erl (**D**).

**Figure 7 ijms-21-01721-f007:**
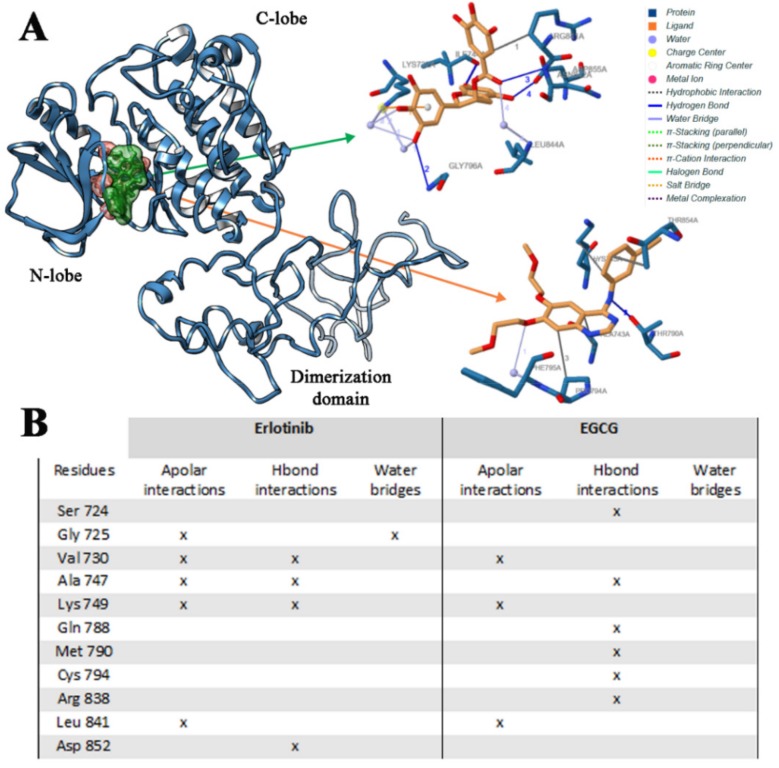
Binding mode of Erlotinib and EGCG in association with wild type EGFR domain (**A**), and interactions list of Erlotinib and EGCG with wild type EGFR (**B**). The 3D binding mode of Erlotinib (orange) and EGCG (green) has been reported.

**Figure 8 ijms-21-01721-f008:**
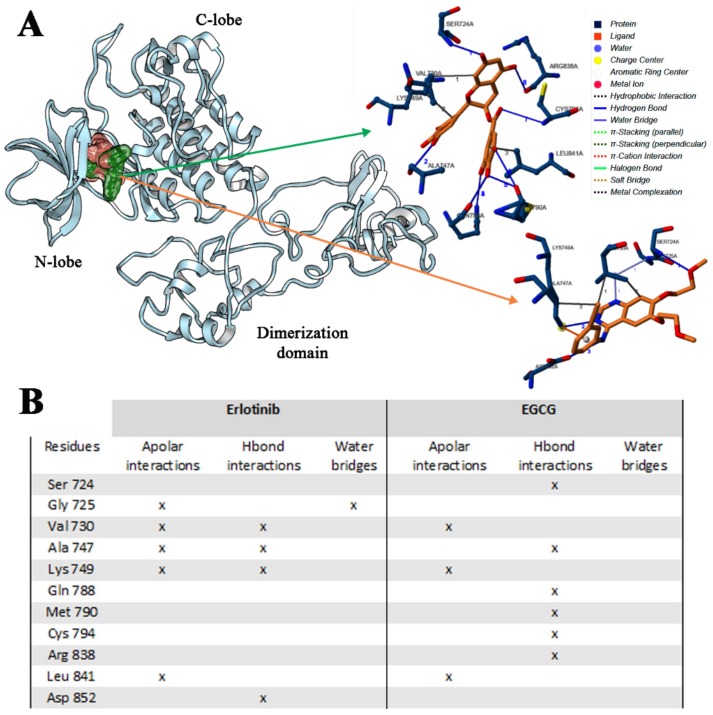
Binding affinity of Erlotinib and EGCG in association with ELREA deletion EGFR domain (**A**), and interactions list of Erlotinib and EGCG with ELREA deletion EGFR (**B**). The 3D binding mode of Erlotinib (orange) and EGCG (green) has been reported.

**Figure 9 ijms-21-01721-f009:**
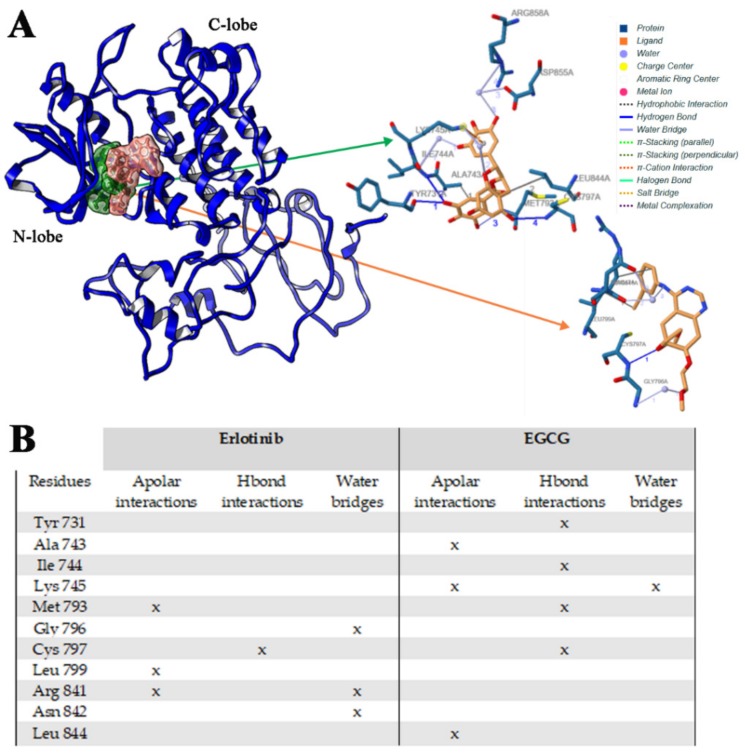
Binding affinity of Erlotinib and EGCG in association with T790M/L858R EGFR domain (**A**), and interactions list of Erlotinib and EGCG with T790M/L858R EGFR (**B**). The 3D binding mode of Erlotinib (orange) and EGCG (green) has been reported.

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
