# Peer review of "Conformational Insight on WT- and Mutated-EGFR Receptor Activation and Inhibition by Epigallocatechin-3-Gallate: Over a Rational Basis for the Design of Selective Non-Small-Cell Lung Anticancer Agents"

_ijms, 2020, doi:10.3390/ijms21051721_

Round 1
Reviewer 1 Report
The manuscript titled “Conformational insight on WT- and mutated-EGFR receptor activation and inhibition by Epigallocatechin-3-gallate: over a rational basis for the design of selective non-small-cell lung anticancer agents” by Minnelli et al., describes a set of computational studies (homology modeling, MD simulations, docking and MM/PBSA simulations) concerning the structure, dynamics and conformations of wt and mutant EGFR, as well as the interactions of these constructs with pharmaceutically relevant inhibitors. EGFR is implicated in difficult to treat non-small cell lung cancer. Overall good agreement was obtained between calculations and experimental findings and it is suggested that the results of the study could be used for the rational design of novel drugs.
Despite the importance of the topic, I have multiple concerns about this manuscript which in my view, prevent its publication, at least in its current form. The main issue I have with this study is the modeling of the protein structure. Since the TK domain of EGFR was not crystallized in its entirety and the dimerization domain was not crystallized at all, this is an extremely challenging modeling task. Yet, no information is provided in the manuscript that would allow estimating how successful it was. For example, no information is provided on the length of the missing loops that had to be modeled (the SI mentions a 29aa loop which is way too long to be correctly modeled), the stereochemical quality of the model was not assessed, template-target RMSD data were not provided and whether or not the model has any resemblance to the real structure is not known. Importantly, if the initial model is inaccurate, I very much doubt whether a 100 ns MD simulation would be enough to correct it. Clearly, without a rigorous evaluation of the quality of the model, the validity of the entire work is questionable.
Another issue has to do with the number of length of the simulations. Each construct was subjected to a single 100 ns MD simulation and the complexes were subjected to single 20 ns simulations. It is advisable to run longer simulations (in particular, a 20 ns simulation is way too short) and to repeat each simulation multiple times.
Finally, two technical issues: (1) The quality of the figures should be greatly improved and (2) The level of English is insufficient to the point where it is sometimes difficult to follow the flow of the manuscript, and should also be improved.
More specific points are listed below:
Line 55: This entire paragraph is unclear. In particular, what is the sensitivity of the WT protein to the different inhibitors and how do the different mutations alter this sensitivity.
Line 57: What are the protein sequence positions of the ELREA mutations?
Line 105: This paragraph deals with the effect of mutations on the equilibrium between the active and inactive forms of the protein. Yet, the importance of this point was not discussed until now, nor its relation (if any) to the sensitivity of the protein or its mutants to the inhibitors.
Line 110: Where is the binding site located?
Line 132: The authors state that they used the 6ewx structure to evaluate the potential dimerization of the WT protein and its mutants. Yet it is unclear how the PDB structure was used in this assessment.
Line 135: It is also not clear whether dimerization studied by and occurred within the course of the MD simulations (unlikely) or whether it was assessed based on the 6ewx structure. If the latter, how relevant is the 6ewx structure for the modeling of EGFR?
Figure 5: An RMSD of about 1Å seems very small, in particular since large parts of the structure were modeled without a relevant template. GROMACS usually reports RMSD values in nm. Are the authors sure about their numbers?
Line 158: It is suggested that the higher fluctuations associated with the T790M/L858R construct are associated with constitutive activation. Yet, in line 113, constitutive activation for this construct was suggested to emerge from additional H-bonds. Wouldn’t one expect such interactions to reduce fluctuations rather than to increase them?
Line 166: What is the advantage of using MSD over RMSD for analyzing the movements of the D domain? Also, if much larger movements were observed for the double mutant and the ELREA constructs, why weren’t these movements observed in the RMSD analysis (Figure 5A)?
Line 175: I don’t think the differences in Rg values between the different constructs are significant. Looking at Figure 5D, there are many fluctuations in this parameter along the simulations and in particular, after 80 ns, the Rg values for all constructs are identical. The subsequent decrease in Rg for the double mutant may well be just a fluctuation.
Line 187: The docking procedure is best described as semi-flexible docking and not as dynamic docking.
Prior to docking, was an attempt made to reproduce the binding mode of erlotinib within the binding site of the 4HJO structure?
Line 189: If the intention is to reorient the ligands inside their binding sites, 20 ns are way too short.
Line 203: The legend of Figure 6(b) is not clear.
Line 212: It is suggested that the MD trajectory did not demonstrate the formation of a stable complex between Erl or EGCG and the T790M/L858R mutant. Does this mean that the complex disintegrated during simulation?
Line 278: Is it possible that the clinically observed inefficient inhibition of T790M/L858R by Erl results from a stronger binding of ATP to this mutant (which makes the competition by the inhibitor more difficult) rather than from a weaker binding of the inhibitor?
Line 295: How does this statement (regarding the combination of EGCG and TKIs) contribute to the discussion?
Line 364: What is the relevance of the docking protocol cited in references 60 and 61 to the present work? The fact that a certain docking protocol performed well on one system does not guarantee that it will perform well on other systems. Details on the MM/PBSA calculations are missing.
SI, line 61: Are you suggesting that RMSD of 0.13±0.05Å is different from an RMSD of 0.12±0.01Å? Such a difference is clearly insignificant.
SI, line 73: Again, are you suggesting that RMSD of 0.13±0.01Å obtained for the binding of Erl to ELREA EGFR is suggestive of a stable conformation whereas an RMSD of 0.18±0.04Å obtained for the binding of EGCG to ELREA EGFR is suggestive of an instable conformation?
SI, line 84: Why is an RMSD of 0.8 Å suggestive of no stabilization pathway? Such small RMSD values simply suggest some structural relaxation and are not indicative in any way of complex stability.
Minor points:
Line 17: A “.” is missing after phosphorylation.
Line 19: Change “clinical” to “clinically”.
Line 21: There is an extra “.” After “EGFR-TKIs”.
Line 23: Change “evidence” to “detect” or something like this.
Line 55: I am not sure “individuate” is the correct word to be used here.
Line 56: What are “drug-caution” mutations?
Line 62: What is “consists on weakens”?
Line 67: Change “tea, known” to “tea, is known”.
Line 68: Change “inhibitor competes” to “inhibitor that competes”.
Line 89: The sentence starting with “it has never been put” should be re-written.
Line 103: Change “We report the same color index on both figures for a major clarity” to “for clarity we report the same color index on both figures”.
Line 113: Change “additive” to “additional”.
Figure 3: What is presented in the “B” panel?
Figure 5D: Why are the units on the X-axis given in ps and not in ns (like in 5A and 5C)?
Line 188: Change “despite they have a” to “despite having a”.
Line 229: “Binding affinity” or “Binding mode”?
Lines 246, 249, 259: Change “respect” to “with respect”.
Line 301: Change “pierce” to “transverses”.
SI, line 61: What is “a relevant fluctuation”?
SI, line 62 and 63: I assume you mean Figures S2A and S2B.
Author Response
REFEREE 1 – point-to point answers
The manuscript titled “Conformational insight on WT- and mutated-EGFR receptor activation and inhibition by Epigallocatechin-3-gallate: over a rational basis for the design of selective non-small-cell lung anticancer agents” by Minnelli et al., describes a set of computational studies (homology modeling, MD simulations, docking and MM/PBSA simulations) concerning the structure, dynamics and conformations of wt and mutant EGFR, as well as the interactions of these constructs with pharmaceutically relevant inhibitors. EGFR is implicated in difficult to treat non-small cell lung cancer. Overall good agreement was obtained between calculations and experimental findings and it is suggested that the results of the study could be used for the rational design of novel drugs.
Despite the importance of the topic, I have multiple concerns about this manuscript which in my view, prevent its publication, at least in its current form. The main issue I have with this study is the modeling of the protein structure. Since the TK domain of EGFR was not crystallized in its entirety and the dimerization domain was not crystallized at all, this is an extremely challenging modeling task. Yet, no information is provided in the manuscript that would allow estimating how successful it was. For example, no information is provided on the length of the missing loops that had to be modelled (the SI mentions a 29aa loop which is way too long to be correctly modelled), the stereochemical quality of the model was not assessed, template-target RMSD data were not provided and whether or not the model has any resemblance to the real structure is not known.
Thanks for your comment. Sorry, we meshed up two parts and we explained wrongly in the methods session.
The tyrosine kinase domain has been largely crystallized; the D-domain domain instead was partially lacking (only half of it has been resolved). Starting from these crystallographic data (see methods section and modified SM), we rebuilt only this part. The mentioned disordered 16 aa loop (from Tyr975 to Asp991) (previously 29aa for a typing error), which is inherently characterized by a high degree of conformational variability, is positioned at the linkage between the ATP-domain and the D-domain (see images) but its coordinates were retrieved from pdb files 5cnn and 4rj4. Thus, since the original configuration was accurate, the subsequent MD simulations can successfully take account for any subsequent motion. The previous mentioned sequence (Pro848-Pro877) in the SM, is retrieved for the pdb template, since it is included as the final part of the TK-crystallized domains. This loop is directly involved in an important aa mutation (Leu858-Thr858), which confers to the EGFR drug resistance and it is important that is present in the pdb original file.
Furthermore, to estimate the goodness of the obtained model, we added a section in the supplementary materials; there, we reported the template-target RMSD value (0.591 Å) and the Ramachandran plot of the entire cytoplasmic EGFR wild type model; it shows that 82 % of residues are in favoured folded region, 14.4% are in additionally allowed region and 4.6% are in outlier region (corresponding to merely to Gly residues). RMSD values and Ramachandran plot proves that we obtained a high local quality estimation. Also Verify 3D results were added.
For all three rebuilt EGFR models, we extrapolated three conformations along 100 ns of MD simulations (derived from cluster analysis) (see figure S3 SM). For each of these conformations, we carried out other 100 ns of molecular dynamics. The aim was to obtain a triplicate of simulations for each model starting from different special dispositions, to evaluate whether there is a convergence in the RMSD values ​​for the TK and dimerization domains. What we have observed is that both of domains reach the same RMSD values along MD simulation, confirming that the reach a convergent final steady state.
Form the detailed analysis, we can confirm that the D domain presents regions with a high disordered degree respect to the TK one, and that the stereochemical quality of EGFR models is assessed.
Importantly, if the initial model is inaccurate, I very much doubt whether a 100 ns MD simulation would be enough to correct it. Clearly, without a rigorous evaluation of the quality of the model, the validity of the entire work is questionable.
Thanks for your comment; we validated the high accuracy degree (as described in the answer to the previous point). Indeed, it was very poorly written, we are sorry for that, since in the methods section, a part was lacking and some mistakes were present. We modelled the core unit of the D-domain using existing templates (4rj4 and 5cnn), while the lacking residues were built using I-TASSER and how specified in the materials section.
More, if the aim would be to accurately reproduce for each model the correct dimerization state, 100 ns MD simulation (even if in triplicate and extended along the MD trajectory for other 100 ns each) could be insufficient. However, our aim was to put into light the evidences that the TK dimerization domain for the double mutant EGFR was actually and permanently in the activated state, thus inducing auto-phosphorylation that could be the reason of its resistance to treatment. For the other considered mutants, we expected a high degree of disorder and oscillation between conformational states, in absence of the natural ligand that through phosphorylation reaction is able to induce a rearrangement of the dimerization domain (see. 0.1021/acs.biochem.5b00444)
Another issue has to do with the number of length of the simulations. Each construct was subjected to a single 100 ns MD simulation and the complexes were subjected to single 20 ns simulations. It is advisable to run longer simulations (in particular, a 20 ns simulation is way too short) and to repeat each simulation multiple times.
Thanks for your comment. Concerning the MD simulation length for the dimerization domain, we have discussed to the previous point.
In relation to receptor inhibitor complexes, our aim was to observe a ligand full accommodation in the ATP binding site after we identified the docking pose. It is worth to remind that 3D structure of this site is located in the crystallized part of the tyrosine kinase domain and thus it is structurally well defined. The general accepted protocol for this dynamic docking, consists in rigid docking of the ligand followed by a brief MD simulation in a range between 10- 20 ns that are sufficient to obtain a ligand better ligand positioning in the binding site, as can be retrieved from literature data. (see for example https://doi.org/10.1016/j.jmgm.2020.107548; https://doi.org/10.1155/2018/3502514).
Finally, two technical issues: (1) The quality of the figures should be greatly improved and (2) The level of English is insufficient to the point where it is sometimes difficult to follow the flow of the manuscript, and should also be improved.
Thanks for your suggestions; we improved both the quality of figures adding original TIFF format files and the level of English.
More specific points are listed below:
Line 55: This entire paragraph is unclear. In particular, what is the sensitivity of the WT protein to the different inhibitors and how do the different mutations alter this sensitivity.
Thanks for your valuable suggestions. We corrected the paragraph to make it clearer; we also added more information in the introduction, in which we specify that EGCG and Erl are small molecules that act as ATP-competitive inhibitors, avoiding its phosphorylation by interacting with the ATP binding site.
The sensitivity of EGFR-protein to Erl is: EGFR-ELREA@ EGFR-wild type >> EGFR-L858R/T790M. In particular between EGFR-wt and the double mutated there is 1600 fold-decrease, IC50, 0.25 vs 413 nM) (doi.org/10.1038/s41598-019-42245-3; doi: 10.1093/annonc/mdt573; 10.1590/1414-431X20144099 ).
Besides, experiments performed with EGCG inhibitor showed approximatively the same growth inhibition effect in the NSCLC cell lines harbouring EGFR-ELREA (doi: 10.18926/AMO/55587; 10.3892/or.2013.2933) and those expressing EGFR-wt. On NSCLC cell lines harbouring EGFR-L858R/T790M the effect is less pronounced. These in vitro data confirmed the in silico results showing a great difference in binding affinity between EGFR-wt and ELREA with the double mutated EGFR-L858R/T790M.
No data are available for direct comparison between Erl and EGCG, since all the experiments were conducted in different conditions, without comparing these two inhibitors.
Line 57: What are the protein sequence positions of the ELREA mutations?
It is Glu746-Ala750 fragment. We added it in the introduction session. (line 64)
Line 105: This paragraph deals with the effect of mutations on the equilibrium between the active and inactive forms of the protein. Yet, the importance of this point was not discussed until now, nor its relation (if any) to the sensitivity of the protein or its mutants to the inhibitors.
We added some periods in the abstract and the introduction section to make it clearer.
Line 110: Where is the binding site located?
As reported in figure 2, the ATP binding site is located in the crystallized part of the TK domain, between C-lobe and N-lobe. We changed the figure caption to make it clear.
Line 132: The authors state that they used the 6ewx structure to evaluate the potential dimerization of the WT protein and its mutants. Yet it is unclear how the PDB structure was used in this assessment.
Thanks for your suggestion, we agree that it was not clearly explained in the text, and we improved the description in the paper.
6ewx refers to a pseudo-kinase and signaling protein Pragmin that contains a classical protein-kinase domain. It has been discovered that this protein uses the tyrosine kinase domain to induce protein tyrosine phosphorylation in human cells. The 3D dimerization structure in its dimeric form (without part of the dimerization domain) has been elucidated and reported in 6ewx, and it has been considered a model for tyrosin-kinase receptors dimerization conformation.
Thus, for all three EGFR forms, we generated the corresponding dimeric structures using the coordinate of the dimeric TK receptor (6ewx). The RMSD value of superimposition of the TK domain was 0.87 Å, 1.01 Å, and 0.98 Å for wild type, T790M/L838R and ELREA deletion forms respectively. As a result, as can be seen in figure 4, for T790M/L838R a matching of the dimerization domains of the two units is observed, thus suggesting it is in its activated state.
Line 135: It is also not clear whether dimerization studied by and occurred within the course of the MD simulations (unlikely) or whether it was assessed based on the 6ewx structure. If the latter, how relevant is the 6ewx structure for the modeling of EGFR?
Thanks for the comment. The 6ewx structure was not used for modeling EGFR, but only to check if the dimerization loop could be in its activated state as it is in the dimeric association. As already stated, the modelling of the dimerization section of the TKdomain has been performed without any restrain.
Figure 5: An RMSD of about 1Å seems very small, in particular since large parts of the structure were modeled without a relevant template. GROMACS usually reports RMSD values in nm. Are the authors sure about their numbers?
Thanks for the suggestion. We were wrong to report the scale for RMSDs related to the protein structures of EGFR, for which the scale of RMSD values ​​are expressed in nm. The scale relative to the RMSD of the ligands were left unchanged, as the RMSD values are expressed in angstroms. We changed the units of measurement in the paper and in the supplementary material.
Line 158: It is suggested that the higher fluctuations associated with the T790M/L858R construct are associated with constitutive activation. Yet, in line 113, constitutive activation for this construct was suggested to emerge from additional H-bonds. Wouldn’t one expect such interactions to reduce fluctuations rather than to increase them?
Thanks for your comment. Constitutive activation is associated with H bonds interactions involving residues 768, 770, 827 and 831, as already known (Ferguson, K.M. A structure-based view of Epidermal Growth Factor Receptor regulation. Annu Rev Biophys. 2008, 37: 353–373); indeed, what we pointed out is the greater fluctuation of some residues of the T790M/L858R form along the MD simulation trajectory that converged in the steady state structure with specific orientation of the D-domain (stable conformation). In fact, this conformational reorientation must be associated with a more marked movement of the residues compared to the other two forms associated with inactive (initial state)-active (MD steady state) conformational switch; this switch didn’t occur for the other two forms that instead keep an intrinsically disordered behaviour.
Line 166: What is the advantage of using MSD over RMSD for analysing the movements of the D domain? Also, if much larger movements were observed for the double mutant and the ELREA constructs, why weren’t these movements observed in the RMSD analysis (Figure 5A)?
Thanks for your suggestion. As reported in Supplementary Material, Mean Square Displacement (MSD) allows quantifying a displacement from a set of initial position. MSD analysis is related only to the dimerization (D) domain, without considering the whole TK domain. The root mean square deviation (RMSD) analysis instead is related to the entire cytoplasmic (TK including D-domain). This tool allows to measure the average distance between the atoms along MD simulation. A low RMSD value for all protein means that our system reaches a steady state but does not quantify the displacement of a part of it.
Line 175: I don’t think the differences in Rg values between the different constructs are significant. Looking at Figure 5D, there are many fluctuations in this parameter along the simulations and in particular, after 80 ns, the Rg values for all constructs are identical. The subsequent decrease in Rg for the double mutant may well be just a fluctuation.
Thanks for suggestion The Rg analysis allows to investigate the compactness degree of the receptor. In this case we have 3 forms of the same receptor, with small amino acid variations, therefore the differences in the Rg values in the order of 1 or more nm represents an important variation in the compactness degree of the protein (DOI:10.1134/S0026893308040195; DOI: 10.1016/S0006-3495(03)74709-2 ).
Line 187: The docking procedure is best described as semi-flexible docking and not as dynamic docking.
Thanks for discussion. Indeed, we applied a dynamic docking approach not a semi-flexible one.
In particular, we carried out molecular docking with protein as rigid structure/the ligand as flexible followed by molecular dynamics simulation, to stabilize bound ligand in its cleft in the physiological condition. This protocol is considered as Dynamic Docking, since it particularly refers to approaches that involve MD simulations either directly (such as in solvent mapping) or indirectly (docking refinement and/or re-scoring). The last one is the one we followed. From this point of view, dynamic docking simulations are distinguished from other approaches (such as merely rigid docking or semi-flexible docking) by the possibility of characterizing the protein-ligand binding process at a fully dynamic level. (see for example Gioia et al. Molecules. 2017 Nov 22;22(11). pii: E2029. doi: 10.3390/molecules22112029; Salmaso et al. Front. Pharmacol., 22 August 2018 | https://doi.org/10.3389/fphar.2018.00923).
Prior to docking, was an attempt made to reproduce the binding mode of erlotinib within the binding site of the 4HJO structure?
Thanks for suggestion. We added some more details to this section. We applied our docking protocol on Erl to reproduce its orientation in 4HJO pdb structure; we obtained the same binding pose found in the crystallographic structure. We already mentioned this in the text (line: 244-245).
Line 189: If the intention is to reorient the ligands inside their binding sites, 20 ns are way too short.
Thanks for comment. We already answered in a previous point. The general accepted protocol for this dynamic docking, consists in rigid docking of the ligand followed by a brief MD simulation in a range between 10- 20 ns that are sufficient to obtain a ligand better ligand positioning in the binding site, as can be retrieved from literature data. (see references in the previous answer)
Line 203: The legend of Figure 6(b) is not clear.
Thanks for comment. We fixed the legend on the text.
Line 212: It is suggested that the MD trajectory did not demonstrate the formation of a stable complex between Erl or EGCG and the T790M/L858R mutant. Does this mean that the complex disintegrated during simulation?
The consideration regarding the poor stability of the complexes with T790M / L858R EGFR is linked to the poor binding affinity of EGCG and Erl in association with this mutated form. The models were intact for all MD simulation, but with binding affinities not comparable with the wild type system, which is known to be inactivated by the two studied ligands.
Line 278: Is it possible that the clinically observed inefficient inhibition of T790M/L858R by Erl results from a stronger binding of ATP to this mutant (which makes the competition by the inhibitor more difficult) rather than from a weaker binding of the inhibitor?
Sure, it should be possible to a little extent. As we observed a decrease in the binding affinity of Erl and EGCG, it is possible that also the binding affinity for ATP has increased, even if the mutations are not directly involving ATP-binding residues and thus we do not expect a high influence. We choose not to investigate the ATP binding mode, neither we evaluated the behaviour of the EGFR-ATP models, since to evaluate any differences in its binding affinity, we would investigate the enzymatic reaction mechanisms that is a long and computationally demanding task.
Line 295: How does this statement (regarding the combination of EGCG and TKIs) contribute to the discussion?
Thanks for the comment. We eliminated the sentence and describe it better in the methodological section (reformulating the paragraphs lines 401-405)
Line 364: What is the relevance of the docking protocol cited in references 60 and 61 to the present work? The fact that a certain docking protocol performed well on one system does not guarantee that it will perform well on other systems. Details on the MM/PBSA calculations are missing.
Thanks for suggestion. We already used this docking protocol for many different computational studies regarding ligand-receptor association. Indeed, it is true that each time, the protocol must be “tuned” to each studied system. However, some basic common features must be used as reported also in the cited works. In this case, we validated it comparing the obtained binding pose Erl/wt-EGFR in the ATP binding site, which is already known to be the unique binding site for the TK inhibitors, with the crystallographic one.
We also added details on MM/PBSA calculations to the methods section.
SI, line 61: Are you suggesting that RMSD of 0.13±0.05Å is different from an RMSD of 0.12±0.01Å? Such a difference is clearly insignificant.
Thanks for your suggestion. In this paragraph, we referred to the estimated error for Erl (0.05Å) respect to that for EGCG (0.01) in the ATP binding site. This means that Erl presents a relevant fluctuation degree respect to EGCG in the ATP binding pocket. We better explained it in the text of Supplementary Material file.
SI, line 73: Again, are you suggesting that RMSD of 0.13 ± 0.01Å obtained for the binding of Erl to ELREA EGFR is suggestive of a stable conformation whereas an RMSD of 0.18±0.04Å obtained for the binding of EGCG to ELREA EGFR is suggestive of an instable conformation?
Thanks for your comment. The answer is in the previous point and we added a description in the text.
SI, line 84: Why is an RMSD of 0.8 Å suggestive of no stabilization pathway? Such small RMSD values simply suggest some structural relaxation and are not indicative in any way of complex stability.
Thanks, you are right. We did not explain it in the proper way. We changes the text. The values are reffered to a DRMSD with respect the initial structure. We meant that during the last 10 ns, there are many fluctuations (DRMSD=0.5-0.8 Å) of its value that is indicative of a poor steady state or also a conformational partial flexible structure.
Minor points:
Line 17: A “.” is missing after phosphorylation.
Done
Line 19: Change “clinical” to “clinically”.
Done
Line 21: There is an extra “.” After “EGFR-TKIs”.
Done
Line 23: Change “evidence” to “detect” or something like this.
Done
Line 55: I am not sure “individuate” is the correct word to be used here.
We replaced “individuate” with “identify”
Line 56: What are “drug-caution” mutations?
Sorry, it was a typing error. It has been substituted with drug-resistance
Line 62: What is “consists on weakens”?
It means that the secondary T790M mutation determines a decrease of binding affinity of canonical inhibitors for ATP binding site. We explained it in the text.
Line 67: Change “tea, known” to “tea, is known”.
Done
Line 68: Change “inhibitor competes” to “inhibitor that competes”.
Done
Line 89: The sentence starting with “it has never been put” should be re-written.
We replaced this sentence with “the EGCG inhibition and the EGFR mutations of the TK domains have never been linked”.
Line 103: Change “We report the same color index on both figures for a major clarity” to “for clarity we report the same color index on both figures”.
Done
Line 113: Change “additive” to “additional”.
Done
Figure 3: What is presented in the “B” panel?
Figure 3 B represents superimposition of T790M/L858R (green) and ELREA deletion (blue) EGFR structures in the corresponding closed conformations.
Figure 5D: Why are the units on the X-axis given in ps and not in ns (like in 5A and 5C)?
Thanks for suggestion, we used the ns unit on X-axis for 5D
Line 188: Change “despite they have a” to “despite having a”.
Done
Line 229: “Binding affinity” or “Binding mode”?
“Binding mode” is more accurate term.
Lines 246, 249, 259: Change “respect” to “with respect”.
Done
Line 301: Change “pierce” to “transverses”.
Done
SI, line 61: What is “a relevant fluctuation”?
The relevant fluctuation is related to the estimated error for the RMSD value of Erl-EGFR model, which is 5 times higher respect to the EGCG-EGFR system. We deleted this expression because we better explained this consideration in the sentences reported in SM.
SI, line 62 and 63: I assume you mean Figures S2A and S2B.
Yes it is.
Reviewer 2 Report
The manuscript by Minnelli et al. explores different computational and in-silico analysis providing insight on WT and mutated EGFR receptor activation. The study is interesting and manuscript is well written. However, there is lack of in-vitro and in-vivo model studies to justfiy the notion established in the analysis. Though the author mentions that "drug design approach aimed to block the mutated EGFR domains activation are currently underway in our laboratory, associating the in silico prediction with in vitro/in vivo experiments to confirm the inhibitory mechanisms against the different EGFR forms and to gain more insights on the evolution and biochemical implication of the molecular structure of the complexes.". Then why not include at least some preliminary results to complement the study in this manuscript. I have reservation for accepting this manuscript until there is proper validation with practical applications through in-vitro or in-vivo studies.
Author Response
REFEREE 2 – answer
The manuscript by Minnelli et al. explores different computational and in-silico analysis providing insight on WT and mutated EGFR receptor activation. The study is interesting and manuscript is well written. However, there is lack of in-vitro and in-vivo model studies to justfiy the notion established in the analysis. Though the author mentions that "drug design approach aimed to block the mutated EGFR domains activation are currently underway in our laboratory, associating the in silico prediction with in vitro/in vivo experiments to confirm the inhibitory mechanisms against the different EGFR forms and to gain more insights on the evolution and biochemical implication of the molecular structure of the complexes.". Then why not include at least some preliminary results to complement the study in this manuscript. I have reservation for accepting this manuscript until there is proper validation with practical applications through in-vitro or in-vivo studies.
Thanks for your comments. We also think that experimental data are very important. Indeed, our study lies on already established model of EGFR activation (refs 22-24 in the paper) and from experimental data available for both Erl and EGCG as inhibitors of EGFR receptors competitive with ATP-binding site (see for example lines 241-252) (refs 19, 32-37). Thus, the aim of this study is to rationalize these experimental findings in order to elucidate the molecular basis of such inhibition.
More experimental details: The sensitivity of EGFR-protein to Erl is: EGFR-ELREA@ EGFR-wild type >> EGFR-L858R/T790M. In particular between EGFR-wt and the double mutated there is 1600 fold-decrease, IC50, 0.25 vs 413 nM) (doi.org/10.1038/s41598-019-42245-3; doi: 10.1093/annonc/mdt573; 10.1590/1414-431X20144099). Besides, experiments performed with EGCG inhibitor showed approximatively the same growth inhibition effect in the NSCLC cell lines harbouring EGFR-ELREA (doi: 10.18926/AMO/55587) and those expressing EGFR-wt. On NSCLC cell lines, harbouring EGFR-L858R/T790M, the effect is less pronounced. These in vitro data confirmed the in silico results showing a great difference in binding affinity between EGFR-wt and ELREA with the double mutated EGFR-L858R/T790M. As a result, we found out a correspondence between in silico and these in vitro data, as pointed out in the discussion section.
Anyway, since the experimental data used are already published, we decided to submit this paper to the Molecular Informatics section of the International Journal of Molecular Sciences, that is more appropiate.
More, starting from this knowledge, we are now designing rationally specific new EGFR inhibitors. A potentially active inhibitor has been found, functionalizing EGCG. Unfortunately, we cannot share our preliminary data since it is under patent.
